

# Statistical response of middle atmosphere composition to solar proton events in WACCM-D simulations: importance of lower ionospheric chemistry

Niilo Kalakoski[1], Pekka T. Verronen[1,2], Annika Seppälä[3], Monika E. Szeląg[1,*], Antti Kero[2], and Daniel R. Marsh[4,5]

[1]Space and Earth Observation Centre, Finnish Meteorological Institute, Helsinki, Finland
[2]Sodankylä Geophysical Observatory, University of Oulu, Sodankylä, Finland
[3]Department of Physics, University of Otago, Dunedin, New Zealand
[4]Atmospheric Chemistry Observations and Modeling, National Center for Atmospheric Research, Boulder, CO, USA
[5]Priestley International Centre for Climate, University of Leeds, Leeds, UK
[*]earlier known as M. E. Andersson

**Correspondence:** Niilo Kalakoski (niilo.kalakoski@fmi.fi)

**Abstract.** Atmospheric effects of solar proton events (SPE) have been studied for decades, because their drastic impact can be used to test our understanding of upper stratospheric and mesospheric chemistry in the polar cap regions. For example, SPEs cause production of odd hydrogen and odd nitrogen, which leads to depletion of ozone in catalytic reactions, such that the effects are easily observed from satellites during the largest events. Until recently, the complexity of the ion chemistry in the lower ionosphere (i.e. in the D region) has restricted global models to simplified parameterizations of chemical impacts induced by energetic particle precipitation (EPP). Because of this restriction, global models have been unable to correctly reproduce some important effects, such as the increase of mesospheric $HNO_3$ or the changes in chlorine species. Here we use simulations from the WACCM-D model, a variant of the Whole Atmosphere Community Climate Model, to study the statistical response of the atmosphere to the 66 largest SPEs that occurred in years 1989–2012. Our model includes a set of D-region ion chemistry, designed for a detailed representation of the atmospheric effects of SPEs and EPP in general. We use superposed epoch analysis to study changes in $O_3$, $HO_x$ ($OH + HO_2$), $Cl_x$ ($Cl + ClO$), $HNO_3$, $NO_x$ ($NO + NO_2$) and $H_2O$. Compared to the standard WACCM which uses an ion chemistry parameterization, WACCM-D produces a larger response in $O_3$ and $NO_x$, weaker response in $HO_x$ and introduces changes in $HNO_3$ and $Cl_x$. These differences between WACCM and WACCM-D highlight the importance of including ion chemistry reactions in models used to study EPP.

## 1 Introduction

A solar proton event (SPE) is a burst of high-energy charged particles, dominated by protons, ejected from the Sun. These protons generally have energies in the range of tens or hundreds of MeVs/nucleon and due to their high energies are able to penetrate deep into the atmosphere at latitudes above $\approx 60°$ magnetic. This causes ionization and dissociation (mainly of the most abundant species $N_2$ and $O_2$) in the altitude range of roughly 30–90 km. Solar proton events typically last from a





few hours to few days and can occur anytime during the 11-year solar cycle, although they are more common during solar maximum.

Ionization and dissociation caused by the SPEs, and energetic particle precipitation (EPP) in general, have a significant influence on neutral composition through ion-neutral chemistry (e.g. Verronen and Lehmann, 2013). Several studies have investigated the depletion of ozone in the polar mesosphere and upper stratosphere resulting from the increased production of odd hydrogen and odd nitrogen (Jackman et al., 2001; Seppälä et al., 2004; López-Puertas et al., 2005; Verronen et al., 2006). Ozone depletion can result in changes in temperatures during the largest SPEs (Jackman et al., 2007), and there is evidence that SPEs, as part of EPP, can modulate polar winter dynamics on decadal time scales (Rozanov et al., 2005; Seppälä et al., 2009; Baumgaertner et al., 2011). Recent results have also shown that SPEs play a role in upper stratospheric variability of ozone and have to be considered when studying the expected recovery of the ozone layer (Stone et al., 2018).

Simulation results have suggested that the decrease of polar total ozone column would be of the order of 1–2% a few months after large SPEs (Jackman et al., 2014), while the contribution of >300 MeV protons to direct ozone loss in the lower stratosphere would likely be negligible due to the relatively small fluxes at such high energies (Jackman et al., 2011). Ozone may also increase in the lower stratosphere due to the enhanced $NO_x$ interfering with chlorine-driven catalytic ozone loss (Jackman et al., 2008). However, local depletion has been reported to reach ≈10% below the ozone layer peak at 50–100 hPa, based on a statistical analysis of almost 200 SPEs using ozone soundings (Denton et al., 2018).

The lower ionospheric, i.e. D-region, chemistry which connects SPE ionization to changes in neutral species is rather complex compared to the 5-ion chemistry that is adequate in the thermosphere. A special feature of the D-region is the presence of negative ions which are formed when electrons attach to neutral species. Another feature is the large number of different cluster ions, both positive and negative, some of which are among the most abundant ions in the region. Due to the large number of D-region ions and ionic reactions, atmospheric models have typically included the SPE, or EPP, effects using simple parameterizations of $HO_x$ and $NO_x$ production.

Funke et al. (2011) discussed shortcomings related to these parameterization schemes in a comparison study between observations and simulations of the October-November 2003 SPE. Among the outstanding issues in simulations have been the lack of nitric acid, $HNO_3$, increase (see also Jackman et al., 2008), as well as lack of chlorine activation. The $HNO_3$ production is driven through reactive nitrogen redistribution by negative ion chemistry (Verronen and Lehmann, 2013), and simulations could be improved either by improved parameterization (Päivärinta et al., 2016), or by considering the relevant ion chemistry explicitly (Verronen et al., 2008; Verronen et al., 2011; Andersson et al., 2016). Difference between modeled and observed response of chlorine species to SPE reported by Jackman et al. (2008) can be explained by the conversion of HCl to active chlorine species (Cl, ClO, HOCl) by the ion chemical reactions (Winkler et al., 2009, 2011). In the lower ionosphere, the ion chemistry reactions are expected to lead to the depletion of water vapour during SPEs. For moderate sized SPEs, this effect is small compared to the icy particle sublimation governed by the changes in temperature (Von Savigny et al., 2007; Winkler et al., 2012).

The Whole Atmosphere Community Climate Model (WACCM) is a global chemistry-climate model and forms the atmospheric part of the Community Earth System Model (CESM). Recently, a variant called WACCM-D was developed for detailed





simulations of D-region ion chemistry and EPP atmospheric effects (Verronen et al., 2016). In a case study of the January 2005 SPE, the consideration of ion chemistry in WACCM-D led to improved SPE response in $HNO_3$, $Cl_x$, $NO_x$, and $HO_x$ (Andersson et al., 2016). Since then, WACCM-D has been used to study mesospheric nitric acid and cluster ion composition during electron precipitation events (Orsolini et al., 2018) and magnetic latitude dependency of SPE ionospheric impact (Heino et al., 2019).

Here, we take a statistical approach to look at the response of middle atmosphere chemistry to a number of SPEs of various intensities. As the SPE effects are largely known from previous work, we focus on the improvement provided by additional ion chemistry reactions included in the WACCM-D chemistry. In addition to the traditionally analyzed species, such as $O_3$, $HO_x$ and $NO_x$, we also investigate water vapour and active chlorine species which have been less studied, especially with global models. While several of the largest SPEs included in the analysis have previously been studied individually, the statistical approach used here allows for inclusion of number of moderate sized events in various background atmosphere and illumination conditions. This approach also allows for identification of robust effects above natural variability. As we analyse SPEs of different sizes occuring during different seasons, a statistical approach is most useful for study of timing and spatial extent, rather than magnitude of the response.

## 2 Modeling and analysis methods

### 2.1 WACCM-D simulations

WACCM is a global circulation model with fully coupled chemistry and dynamics extending from surface to $6 \times 10^{-6}$ hPa ($\approx$140 km). The horizontal resolution used is 1.9° latitude by 2.5° longitude. A description of the model physics in the MLT (mesosphere-lower thermosphere) as well as the simulations of dynamical and chemical response to radiative and geomagnetic forcing during solar maximum and minimum are described by Marsh et al. (2007). Marsh et al. (2013) presents an overview of the model climate and describes the climate and the variability in long-term simulation using version 4 of the WACCM. The standard chemistry package of WACCM includes photochemistry of 59 neutral species for the whole altitude range, and five ion species $O^+$, $NO^+$, $O_2^+$, $N_2^+$ and $N^+$ for the lower thermosphere. For SPE effects, and energetic particle precipitation in general, $HO_x$ and $NO_x$ production is parameterized. For the SPE $NO_x$ effect, it is assumed that 1.25 N atoms are produced per ion pair with branching ratios of 0.55/0.7 for $N(^4S)/N(^2D)$, respectively (Jackman et al., 2005; Porter et al., 1976). This parameterization depends on a fundamental assumption of fixed $N_2$ / $O_2$ ratio, and it has been shown to underestimate $NO_x$ production above about 65 km (Nieder et al., 2014; Andersson et al., 2016). For the SPE $HO_x$ effect, based on the work of Solomon et al. (1981), a maximum of two molecules are produced per ion pair in the lower mesosphere and below while in the upper mesosphere the production gradually decreases to zero with increasing altitude.

WACCM-D is a variant of WACCM in which the standard parameterizations of $HO_x$ and $NO_x$ production are replaced by a set of lower ionospheric photochemistry, with the aim to reproduce better the observed effects of EPP on the mesosphere and upper stratosphere neutral composition. The ion chemistry set was selected based on analysis of the latest knowledge of chemical reactions in the ionospheric D-region and their effects on the neutral atmosphere (Verronen and Lehmann, 2013).



The set consists of 307 reactions of 20 positive ions and 21 negative ions, including cluster ions such as $H^+(H_2O)$ and $NO_3^-(HNO_3)$ which are important for, e.g., $HO_x$ and $HNO_3$ production. The details on WACCM-D ion chemistry as well as its lower ionospheric evaluation were presented by Verronen et al. (2016), while the improvement in the atmospheric response during the January 2005 SPE was demonstrated in comparisons with satellite observations (Andersson et al., 2016). WACCM-D is now included as an official predefined component set (compset) since the CESM 2.0 release in June 2018.

In this study, we use WACCM version 4 simulations with the specified dynamics configuration (SD-WACCM) which is forced with meteorological fields (temperature, horizontal winds, and surface pressure) from NASA GMAO GEOS5.1 (Reinecker et al., 2008) at every dynamics time step below about 50 km; above this, the model is free running (88 levels in total). It should be noted that even above 50 km model dynamics are strongly modulated by winds and wave fluxes from lower levels. This means that internal variability of dynamics in SD-WACCM is small.

For energetic particle precipitation, the simulations include forcing from auroral electrons ($E < 10$ keV), solar protons, and galactic cosmic rays. Medium-energy and high-energy electrons ($E > 10$ keV) were excluded from the simulations. Two model simulations were made: (1) a reference run using standard SD-WACCM compset (referred to as REF) and (2) a run with D-region ion chemistry (referred to as WD). Both simulations covered the years 1989 to 2012. We use daily mean SPE ionization rates based on GOES (Geostationary Operational Environmental Satellite) observations and described, e.g., in Jackman et al. (2011). The energy range for protons is 1–300 MeV, thus the direct atmospheric impact takes place at altitudes above ≈10 hPa (e,g, Turunen et al., 2009, Figure 3).

Proton fluxes were only included for energies up to 300 MeV. However, Jackman et al. (2011) show that higher energies contribute very little to the total ozone or $NO_y$ impact. We only consider protons, and not the X-ray flares that are associated with some SPEs - in the polar regions we can assume that the proton effect is the dominant one, at least for large SPEs (Pettit et al., 2018).

## 2.2 Analysis methods

For our statistical analysis, SPEs were selected using proton flux data from satellite-based GOES observations (available from https://www.ngdc.noaa.gov/stp/satellite/goes/index.html). An event was selected if the peak proton flux exceeded 100 particle flux units (pfu), with pfu defined as the five-minute average flux in units of particles $cm^{-2}s^{-1}sr^{-1}$ for protons with energy larger than 10 MeV. To avoid obscuring the signal in composite means by introducing duplicates of major events in days preceding the zero epoch date, any event following a larger event onset within seven days was excluded from the analysis. In total, eleven such events were removed. Following these criteria, 66 events given in Table A1 were identified in the simulation period. The seasonal distribution of the selected SPEs is shown in Figure 1. As SPEs are sporadic, the distribution is somewhat uneven, with the northern hemisphere winter months (Dec – Feb) underpresented. The largest SPEs with proton fluxes over $10^4$ pfu are focused towards the end of year (Oct – Nov).

For each of the 66 events, zero epoch day ($D_0$) was defined as the first day when the proton flux exceeded 10 pfu. A 90-day epoch period around the zero epoch ($[D_0 - 29, D_0 + 60]$) was selected from the daily mean time-series for the following





**Table 1.** Definitions used in the text for the SD-WACCM and WACCM-D simulations.

| | |
|---|---|
| REF | Daily mixing ratios from the standard SD-WACCM simulation. |
| $\overline{REF}$ | Daily climatological (1989-2012) mean mixing ratios from the standard SD-WACCM simulation. |
| $\widehat{REF} = REF - \overline{REF}$ | Daily mixing ratio anomalies from the standard SD-WACCM simulation. |
| WD | Daily mixing ratios from the WACCM-D simulation. |
| $\overline{WD}$ | Daily climatological (1989-2012) mean mixing ratios from the WACCM-D simulation. |
| $\widehat{WD} = WD - \overline{WD}$ | Daily mixing ratio anomalies from the WACCM-D simulation. |
| Composite mean | Mean over the 66 individual cases. |

constituents: $O_3$, $HO_x$ ($OH + HO_2$), $Cl_x$ ($Cl + ClO$), $HNO_3$, $NO_x$ ($NO + NO_2$) and $H_2O$. Definitions of abbreviations used in the following text for the statistical quantities from REF and WD simulations are shown in table 1.

To assess the anomalies $\widehat{REF}$ and $\widehat{WD}$ associated with the SPEs, daily climatological mean mixing ratios $\overline{REF}$ and $\overline{WD}$ were deducted from respective epoch mixing ratios REF and WD. Figure 2 shows the composite means of $\overline{WD}$ for northern polar cap (60–90 °N, area-weighted). Note that due to the seasonal distribution of the events some seasonal signals remain visible in the composite means, e.g. slight increase of mesospheric $O_3$ during the 90-day epoch period. It can also been seen that for some constituents, especially $HNO_3$, mixing ratio enhancements following the events are large enough to visibly affect the daily climatology.

For each constituent, we calculate the composite mean of difference $\mu_d$ between the epoch time series and the corresponding climatological time series

$$\mu_d = \frac{\sum_{i=1}^{66}(r(t_i) - c(t_i))}{66}, \quad t_i = D_{0_i} + d,$$

where $r$ is the relevant mixing ratio epoch time series REF or WD, $c$ is the epoch time series from daily climatology $\overline{REF}$ or $\overline{WD}$, and $d$ the number of days before and after zero epoch $D_{0_i}$. Subtracting the climatological values from the epoch time series helps to separate the SPE signal from the seasonal signals which can arise from the uneven monthly distribution of the SPEs (see Figure 1). Variance of the composite means was approximated by bootstrapping, i.e. by re-sampling the random selection of dates 1000 times with replacement, with N = 66 for each sample. Bootstrapping was chosen for variance estimation due to its independence of the shape of the distribution, i.e. it does not require a normal distribution. The standard deviation of the variance was then used to identify robust responses, i.e. SPE-driven changes that are clearly larger than the normal variability.

The SPE-driven anomalies are expected to occur following the SPE onset (zero epoch). However, in Section 3 we will show that the results sometimes have also anomalies that extend over the full epoch period. These anomalies are in most cases related to solar cycle variability, due to the SPE sampling over-representing the solar cycle maximum years when compared to climatology. Composite mean SPE ionization for the analyzed epochs is shown in Figure 3. In addition to main ionization peak within ca. 5 days following the onset of the SPE, secondary ionization peaks are evident in the epoch mean, due to closely





separated SPEs. While these secondary peaks are much weaker (by roughly an order of magnitude) than the main peak, they can still cause noticeable anomalies in some of the analyzed constituents.

In order to evaluate the contribution of the improved ion chemistry of WACCM-D, the same epoch analysis was made for both the REF and the WD simulations. The anomalies from the WACCM-D simulation ($\widehat{WD}$) were then compared to anomalies

from the standard WACCM ($\widehat{REF}$). We considered the difference of composite means of differences in mixing ratio between the two simulations for full altitude range (Figures 6 and 7) as well as comparison of the SPE response at pressure levels chosen for maximum difference between two simulations (Figures 8 and 9).

## 3  Results and discussion

### 3.1  Statistical response from WACCM-D

Figures 4 (Northern Hemisphere, NH) and 5 (Southern Hemisphere, SH) show the mean anomaly of the super-imposed epochs for $O_3$, $HO_x$, $Cl_x$, $HNO_3$, $NO_x$ and $H_2O$. Statistically robust anomalies, i.e. those larger than two times the standard deviation of the bootstrap variance, are shown in color and discussed below.

Several constituents show the most pronounced anomaly immediately following the event onset (zero epoch) which in the figures is marked with a black vertical line. For some constituents, most notably $H_2O$ and $Cl_x$, persistent anomalies extending

throughout the epoch period are seen as well. These anomalies can be considered to represent the solar cycle signal because the distribution of SPEs concentrates on the years around the solar maximum. Some anomalies are clearly affected by the response to closely separated SPEs, i.e. when the separation is smaller than 60 days. These are most notable in NH around days $-15$, $+22$ and $+45$ wrt. to zero epoch, corresponding to the secondary peaks of SPE ionization (see Figure 3).

Ozone (top-left panel, Figures 4 and 5) decreases between 1 and 0.01 hPa immediately following the SPE, lasting about

5–10 days. This is caused by catalytic destruction from enhanced $HO_x$. Ozone loss also takes place around 1 hPa, driven by $NO_x$ and $Cl_x$. The depletion around 1 hPa lasts much longer, driven by the $NO_x$ increase, with maximum ozone decrease seen about 30 days after the event onset. In contrast, an increase of ozone is seen throughout the period near the secondary ozone maximum above 0.01 hPa, more robustly in the southern hemisphere. This increase is linked to enhanced atomic oxygen production by $O_2$ photolysis in solar maximum conditions (see also Marsh et al., 2007). In the stratosphere below 5 hPa,

the ozone response is not consistently robust, although increase in this region is seen throughout the epoch period. Thus the decrease of total ozone column, as reported by Denton et al. (2018), is not seen in our analysis. It should be noted that the highest energy protons ($E > 300$ MeV), that can directly impact the lower stratosphere, are not included in the particle forcing used in our simulations. However, it is not clear if inclusion of these protons would change the response because their fluxes should be too low to cause a significant extra response (Jackman et al., 2011).

Strong, short-lived increase of $HO_x$ is seen below 0.01 hPa (top-center panels) immediately following the SPE, reaching down to the 10 hPa level, with strongest response in the mesosphere. Due to the short chemical lifetime of $HO_x$, there is no longer-term anomaly beyond 10 days from the SPE onset. The later peaks just below 0.01 hPa are caused by SPEs occurring within 60 days of zero epoch of analysed SPE. Above 0.01 hPa, a small negative anomaly is present throughout the period,





related to solar maximum through less water vapour in the region and lowering of the $HO_x$ peak altitude. The persistent positive anomaly at just below 0.01 hPa, seen in the SH, is consistent with lowering of the $HO_x$ peak altitude.

$Cl_x$ mixing ratios (top-right panels) are enhanced between 1 and 0.01 hPa, these SPE-driven increases last up to about a week. Below 1 hPa, at the altitude of $Cl_x$ mixing ratio maximum, $Cl_x$ is a reduced. This decrease is roughly consistent

throughout the epoch period, so is likely, at least in part, connected to the solar cycle rather than individual events. However, the magnitude of the negative anomaly increases following the onset of the SPE, with maximum value in the SH reached after 30 days. This effect is qualitatively consistent with short-lived decrease of the NH polar upper stratosphere ClO during the January 2005 SPE as reported both in satellite observations and models (Damiani et al., 2012; Andersson et al., 2016). Here we look at the polar average, but it should be noted that in the presence of SPE-generated $HO_x$, response of $Cl_x$ in the

polar atmosphere depends strongly on the sunlight conditions and varies across latitudes and altitudes (Funke et al., 2011). For example, in polar night stratosphere ClO is converted to HOCl in reaction with $HO_2$, while in mesosphere HOCl is converted to ClO by OH.

Large increase in $HNO_3$ (bottom-left panels) is seen up to 10 days after the SPE onset, with a maximum between 1–0.01 hPa. The absence of a longer-term effect is due to the fast photodissociation of $HNO_3$ in sunlit conditions. Similar to $HO_x$, the later

peaks around days 20 and 45 in the NH are caused by nearly coincident SPEs. Longer-term enhancement persists for 20–30 days after SPEs below 1 hPa. The anomalies are weaker in the SH than in the NH, due to the strongest SPEs mostly occurring during NH winter (see Figure 1) when there is less solar radiation and $HNO_3$ photodissociation.

SPE-driven enhancements of $NO_x$ (bottom-center panels) reach down to 1 hPa level and below. These enhancements start to diminish after about 10 days, but robust longer-term enhancements can be seen for several weeks afterwards. This is especially

clear at lower altitudes (around 1 hPa), where transport from above contributes to persistence of enhancements. Response is different in the northern and southern hemispheres due to the seasonal distribution of the events. In the NH, several strong, closely separated winter-time events show up as recurring anomalies, which are then transported to lower altitudes. The SH response is less variable, with small but robust enhancements seen through the two months following the SPE onset. In addition to the SPE-driven anomaly, there is a persistent, underlying anomaly seen above about 10 hPa from enhanced $NO_x$ production

in the lower thermosphere by solar EUV and EPP during solar maximum times (see also Marsh et al., 2007).

During an SPE, $H_2O$ would be expected to decrease when it is converted to $HO_x$ in ionic reactions. In our analysis, $H_2O$ (bottom-right panels) shows consistent and statistically robust negative anomaly throughout the period analysed, except in the SH lower stratosphere. There is some indication of a stronger anomaly following the SPEs above 0.1 hPa, i.e. at altitudes where $HO_x$ increases. But in general the anomaly is mostly related to the solar cycle through the increased Lyman-$\alpha$ photodissociation

during solar maximum decreasing $H_2O$ in the mesosphere-lower thermosphere (see also Schmidt et al., 2006; Marsh et al., 2007).

### 3.2 Effects of D-region ion chemistry

Figures 6 and 7 show the difference between epoch responses in the NH and SH, respectively, from WACCM-D and the standard WACCM. As the specified dynamics of the two simulations are identical, differences seen in these figures arise from



the addition of the D-region ion chemistry. Note that the robustness threshold for these figures is set to one standard deviation rather than the two standard deviations used in Figures 4 and 5. The differences shown in figures for constituents other than $O_3$ and $H_2O$ also satisfy the two standard deviation threshold. The effects on $O_3$ and $H_2O$, however, do not reach the two standard deviation threshold. Further, Figures 8 and 9 show line plots of epoch response from both WACCM-D ($\widehat{WD}$) and standard

WACCM ($\widehat{REF}$) at pressure levels corresponding to maximum significant difference in Figures 6 and 7.

Ozone (top-left panels) depletion is larger in WACCM-D for about 5 days after the onset at around 0.5 hPa. This effect is very similar in both hemispheres. Both the duration and altitude range of this extra ozone loss clearly correspond to enhanced $Cl_x$, while there is clear no connection to $NO_x$ or $HO_x$ differences. In the SH, less ozone depletion takes place in WACCM-D around 0.01 hPa level, connected to the clearly lower $HO_x$ enhancement in WACCM-D. The reduced $HO_x$ response in

WACCM-D is a result of the lower water vapor amount, and thus lower $HO_x$ production, than what is assumed when $HO_x$ production is parameterized (Andersson et al., 2016).

$Cl_x$ is enhanced in both hemispheres above 1 hPa in WACCM-D, while there is only a relatively weak increase in the standard WACCM. The reason for the enhancement are the ion reactions included in WACCM-D that convert more HCl to active chlorine species Cl and ClO (Winkler et al., 2009) than the neutral gas-phase reactions which are included in both

standard WACCM and WACCM-D. The negative anomaly of $Cl_x$ below 1 hPa seen in Figures 4 and 5 is not present here, again implying that this anomaly is not predominantly due to ion chemistry.

The $HNO_3$ enhancement is clear and strong in WACCM-D, as ion chemistry produces it from other $NO_y$ species. Standard WACCM, like other models using an EPP lookup table parameterization, is known to underestimate the $HNO_3$ mixing ratios when compared to observations (Jackman et al., 2008; Funke et al., 2011; Andersson et al., 2016). A discrepancy between

WACCM-D and standard WACCM $HNO_3$ enhancements is clearly seen in Figures 8 and 9. Due to the dramatic increase on $HNO_3$ following SPEs, peaks from other, close-by events are also clearly seen in Figure 8.

In the mesosphere, the $NO_x$ enhancement is much larger in WACCM-D than in standard WACCM. Below 0.1 hPa there are some signs of an increase in transported $NO_x$, although this effect is not statistically robust. Increased $NO_x$ is consistent between the hemispheres. Andersson et al. (2016) attributes the increased $NO_x$ to enhanced production above 80 km (ca.

0.01 hPa), mostly due to the reaction $O_2^+ + N_2 \rightarrow NO^+ + NO$, and consequent transport to lower altitudes.

Differences in $H_2O$ anomalies (Figures 6–9, lower-right panels) between WACCM-D and the standard WACCM are seen above 1.0 hPa but they are almost entirely overcome by the variance, which indicates that negative anomalies in Figures 4 and 5 are not primarily connected to ion chemistry reactions but are a solar cycle signal. However, a short-lived negative difference , robust to $1\sigma-$ level, can be seen at 0.01 hPa immediately following the onset of SPE, even though the co-located

$HO_x$ production in WACCM-D is smaller. This negative difference is more clearly seen in Figures 8 and 9, where we see that mesospheric $H_2O$ anomalies are very consistent between the simulations, except within 10-15 days following the onset of SPE, where the WACCM-D negative anomalies are larger.



### 3.3 Effect from individual events

The SPE effects are dependent on the background atmosphere and season of the year. Here we discuss this by looking at effects of individual events at selected altitudes. This is also an additional check on the robustness of our analysis.

Figure 10 shows the individual timeseries of differences of epochs from the climatology around 0.05 hPa pressure level
(calculated as the mean of three pressure levels centered at 0.04 hPa). SPEs used in this figure are screened for closely separated events, i.e. events which are followed by a larger event within seven days are not shown. Individual events are sorted from top to bottom by the maximum observed proton flux, with strongest events on top of the figure. Dashed horizontal lines show the cut-offs for 1000 pfu (top line) and 100 pfu (bottom line) SPEs.

As expected, the largest SPEs cause astronger response in $O_3$, $HO_x$ and $NO_x$. The response to events is more pronounced
in the NH due to the seasonal distribution of SPEs, i.e. there are more events in NH winter. This is especially apparent for $NO_x$, as its lifetime is heavily influenced by the amount of sunlight available. On visual inspection, the response is dominated by the largest SPEs, with anomalies following the onset of SPEs being visually indistinguishable for weakest events. Below the 100 pfu threshold, any anomalies following the event onset are dominated by larger events occurring within the analysis period, rather than by the analysed event itself. Thus, the decision to only analyse larger events seems reasonable because the
events smaller than 100 pfu would only add a substantial amount of noise to the analysis.

### 4 Conclusions

We present an analysis of the chemical impacts of SPE on the middle atmosphere using simulations from WACCM-D, a variant of the Whole Atmosphere Community Climate Model including a set of D-region ion chemistry reactions. The aim of our analysis is to study the impact and improvement from detailed ion chemistry scheme, which is done by comparing
WACCM-D results to those from the standard WACCM scheme which used a simple parameterization of $HO_x$ and $NO_x$ production. Instead of analysing individual SPE events, we present a statistical analysis of the 66 largest SPEs over the years 1989–2012 to allow for more general conclusions in a range of different background atmospheric conditions and with the observed seasonal distribution of SPEs. This statistical approach allowed for the incorporation of smaller events which would have been difficult to analyze separately. Analysis was performed by means of superposed epoch method with bootstrapping
to identify statistically robust responses.

Statistically robust SPE signals were present in WACCM-D for all analyzed species ($O_3$, $HO_x$, $Cl_x$, $HNO_3$, $NO_x$), except water vapour. These signals are qualitatively in line with previously published modelling results and satellite observations considering individual events. In addition, our analysis shows longer term, solar cycle type signals for several species due to SPEs predominantly occurring during solar maximum years. In general, the responses are consistent between the two hemispheres.
Compared to the standard WACCM, WACCM-D provides solution to some known shortcomings in chemistry-climate models, particularly in the case of $HNO_3$ and $Cl_x$ response to SPEs. Ozone loss following the SPEs was enhanced at 0.5 hPa in WACCM-D. As there is no corresponding increase of $HO_x$ or $NO_x$ at this altitude, this loss is connected to increased conversion of HCl to reactive $Cl_x$ species by the ion reactions (Winkler et al., 2009), now accounted for in WACCM-D. Conversely,





less ozone loss took place around 0.01 hPa in the SH, due the less production of $HO_x$, in agreement with Andersson et al. (2016). $NO_x$ production in the mesosphere is enhanced by the inclusion of ion chemistry and subsequent downward transport extending the effect to lower altitudes for up to 20 days following the SPE onset. Ion chemistry was found to be crucial for characterization of $HNO_3$ production following the SPEs which confirms the previous results from 1-D chemistry models

(Verronen et al., 2008; Verronen et al., 2011). Some effect was also found in $H_2O$ in an altitude region near the mesopause. This, however, has a large 11-year solar cycle signal, partially masking the SPE impact.

In summary, incomplete representation of EPP-driven chemistry in simulations, for example regarding the $HNO_3$, has been recognized as one of the outstanding questions in understanding the solar influence in middle atmosphere and below (Jackman et al., 2008; Funke et al., 2011). Our results clearly show the importance of D-region ion chemistry in capturing the effects

of energetic particle precipitation in the mesosphere-lower thermosphere simulations. Improved global modeling with ion chemistry such as in WACCM-D, provides an important tool for interpretation of wider range of satellite-based observations of neutral species, and allows for global studies of ionospheric D region.

*Code and data availability.*    All model data used are available from corresponding author by request (niilo.kalakoski@fmi.fi). CESM source code is distributed freely through a public subversion code repository (http://www.cesm.ucar.edu/models/cesm1.0/)

*Competing interests.*    Authors declare that no competing interests are present.

*Acknowledgements.*    A.K. is funded by the Tenure Track Project in Radio Science at Sodankylä Geophysical Observatory. D.R.M. was supported in part by NASA grant NNX12AD04G. The National Center for Atmospheric Research is operated by the University Corporation for Atmospheric Research under sponsorship of the National Science Foundation. Work was carried out as a part of International Space Science Institute (ISSI) project "Space Weather Induced Direct Ionisation Effects On The Ozone Layer"





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

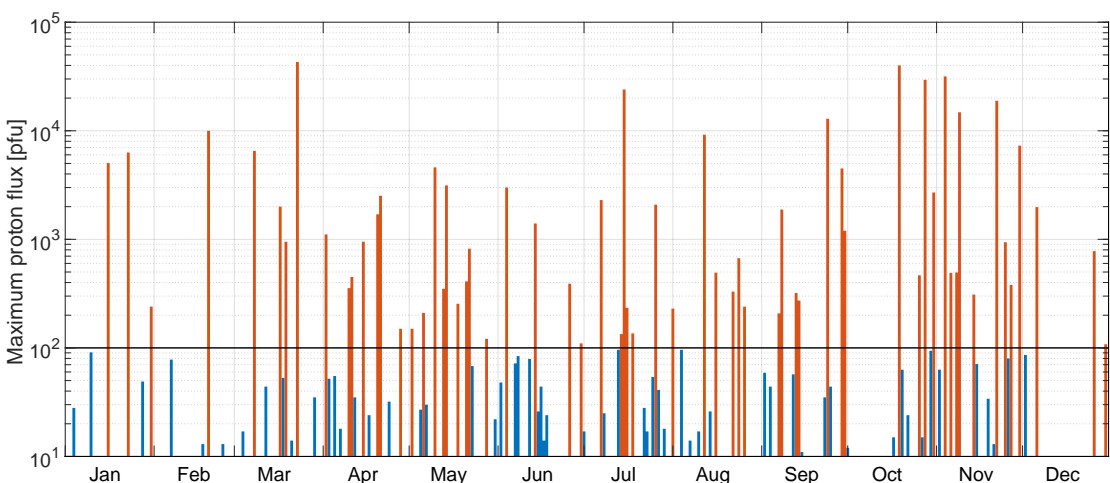

**Figure 1.** Maximum proton flux of solar proton events used in the analysis as a function of day-of-year of the SPE onset. Events with maximum flux over threshold value (100 pfu) are shown as red bars and smaller events as blue bars.

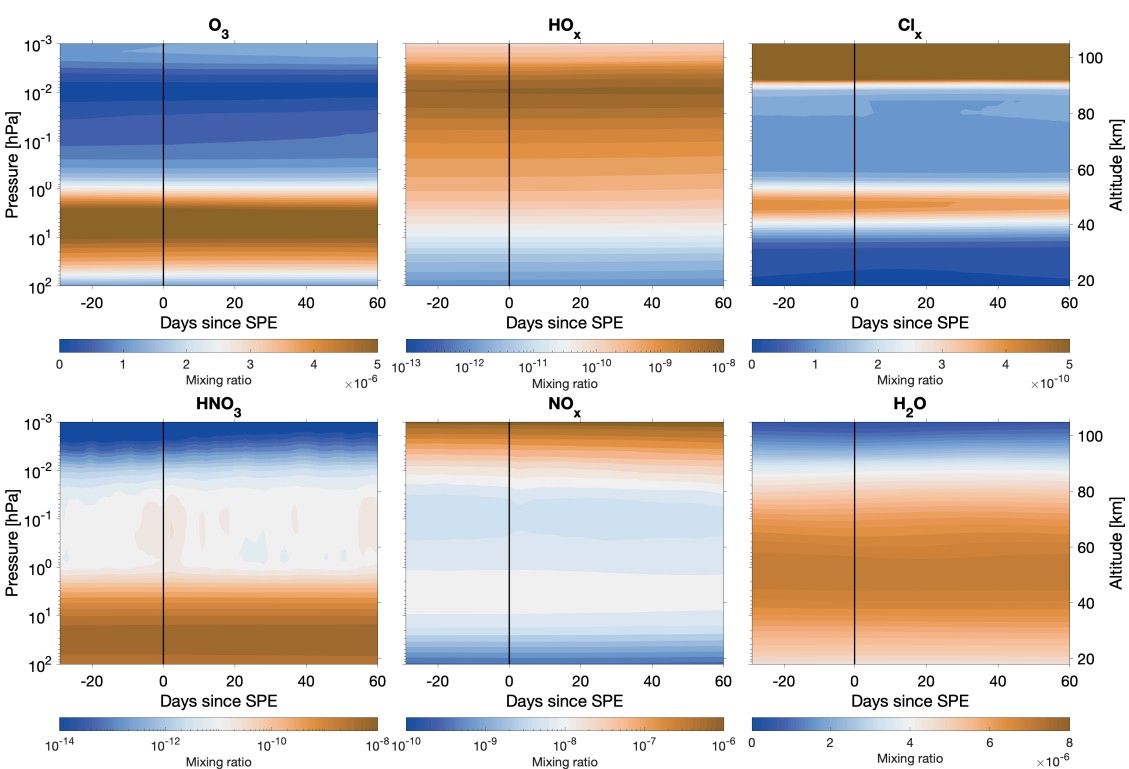

**Figure 2.** Area-weighted northern polar cap (latitude $> 60°N$) average of composite mean of $\overline{WD}$ for $O_3$, $HO_x$, $Cl_x$, $HNO_3$, $NO_x$ and $H_2O$. X-axis shows the number of days before and after the event (day 0, solid black line) and y-axis pressure levels in model (hPa, left) and approximate altitude (km, right).



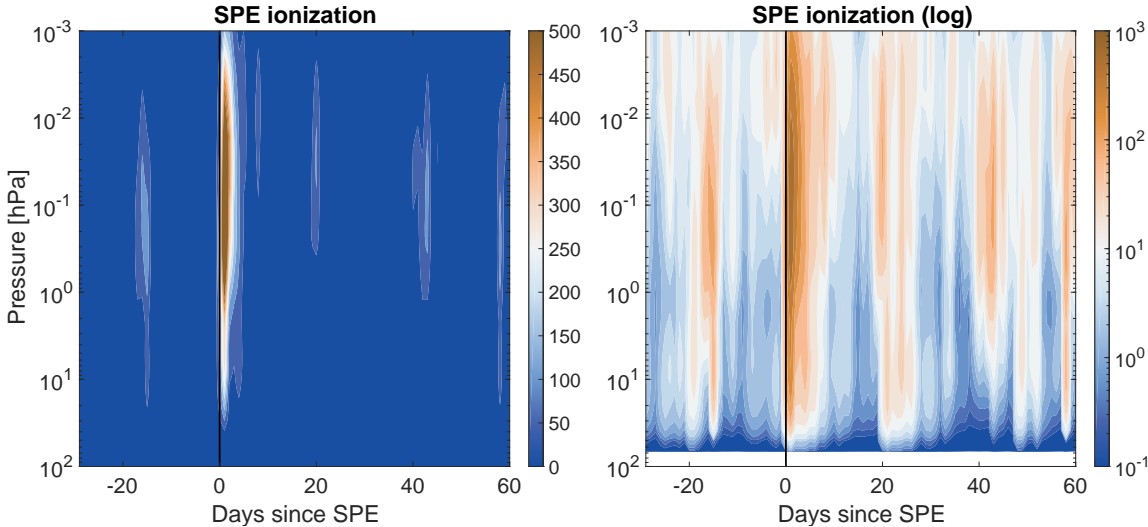

**Figure 3.** Composite mean of epoch time series of SPE ionization rates in linear (top) and logarithmic (bottom) color scale. X-axis shows the number of days before and after the event (day 0, solid black line) and y-axis pressure levels in model.

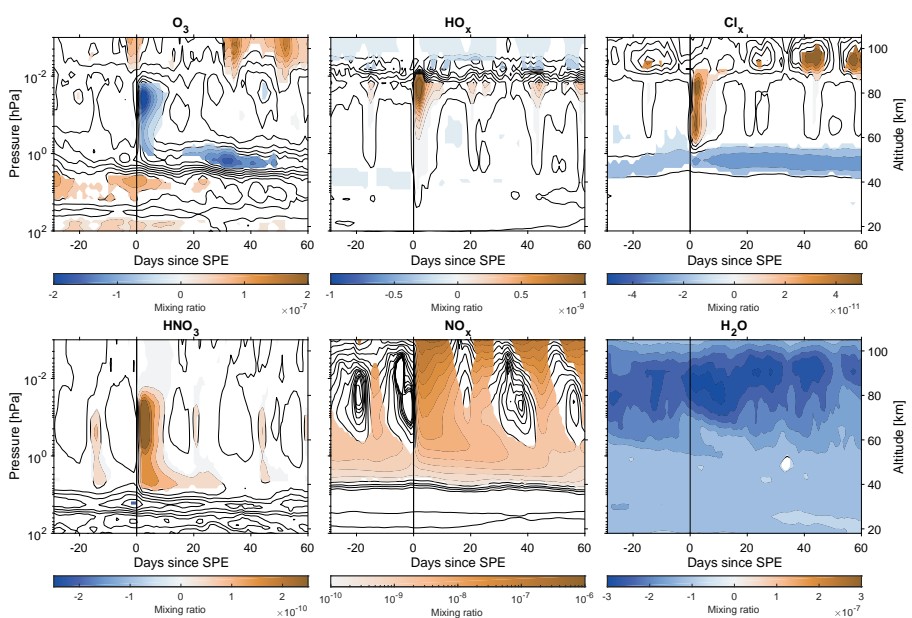

**Figure 4.** Area-weighted northern polar cap averages (latitude $> 60°$N) of composite means of $\widehat{\mathrm{WD}}$ for $O_3$, $HO_x$, $Cl_x$, $HNO_3$, $NO_x$ and $H_2O$. X-axis shows the number of days before and after the event (day 0, solid black line) and y-axis pressure levels in model (hPa, left) and approximate altitude (km, right). Note the logarithmic color scale for $NO_x$.

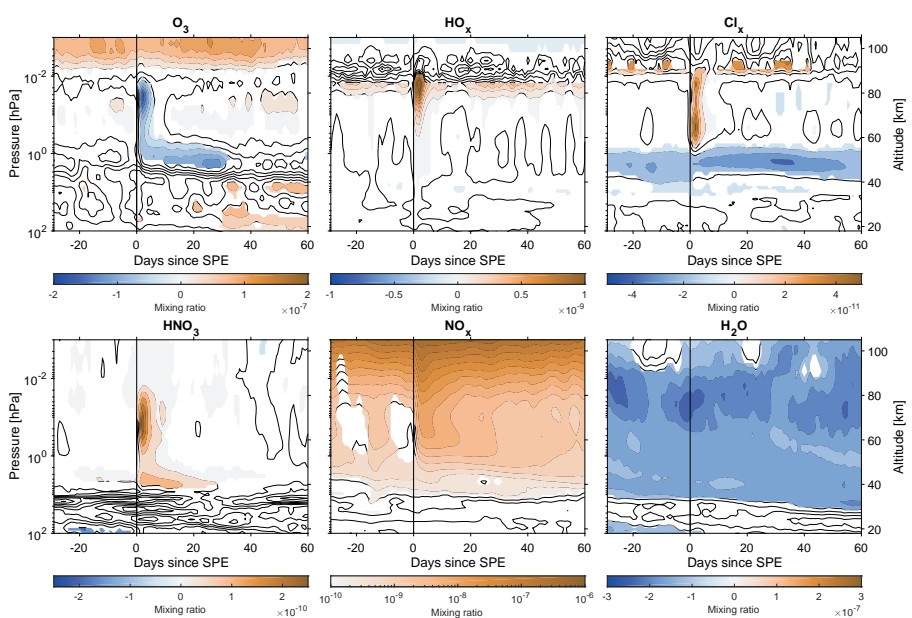

**Figure 5.** Area-weighted southern polar cap averages (latitude > 60°S) of composite means of $\widehat{WD}$ for $O_3$, $HO_x$, $Cl_x$, $HNO_3$, $NO_x$ and $H_2O$. X-axis shows the number of days before and after the event (day 0, solid black line) and y-axis pressure levels in model (hPa, left) and approximate altitude (km, right). Note the logarithmic color scale for $NO_x$.



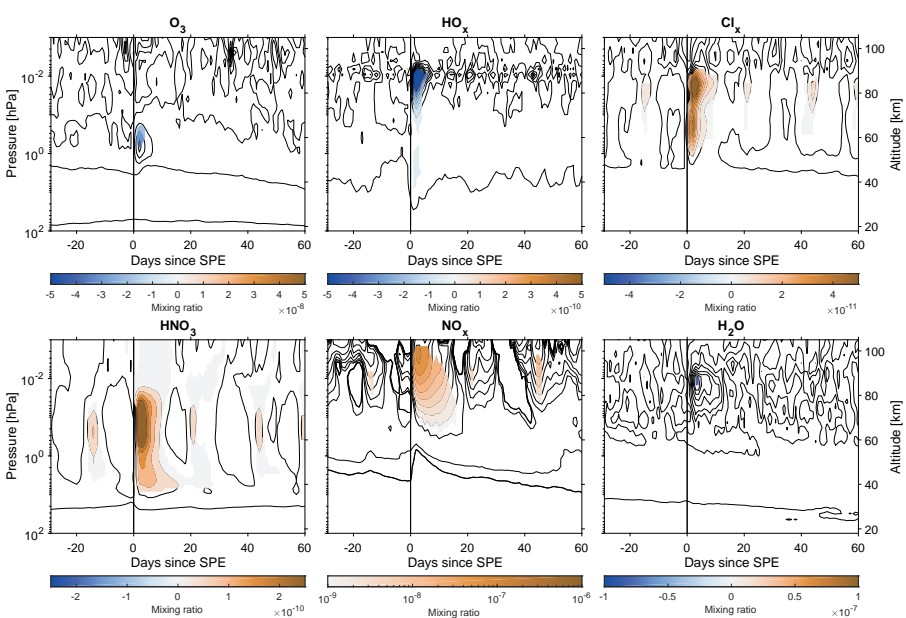

**Figure 6.** Area-weighted northern polar cap averages (latitude $> 60°N$) of composite means of $\widehat{WD} - \widehat{REF}$ (see figure 4). X-axis shows the number of days before and after the event (day 0, solid black line) and y-axis pressure levels in model (hPa, left) and approximate altitude (km, right). Note the logarithmic color scale for $NO_x$.

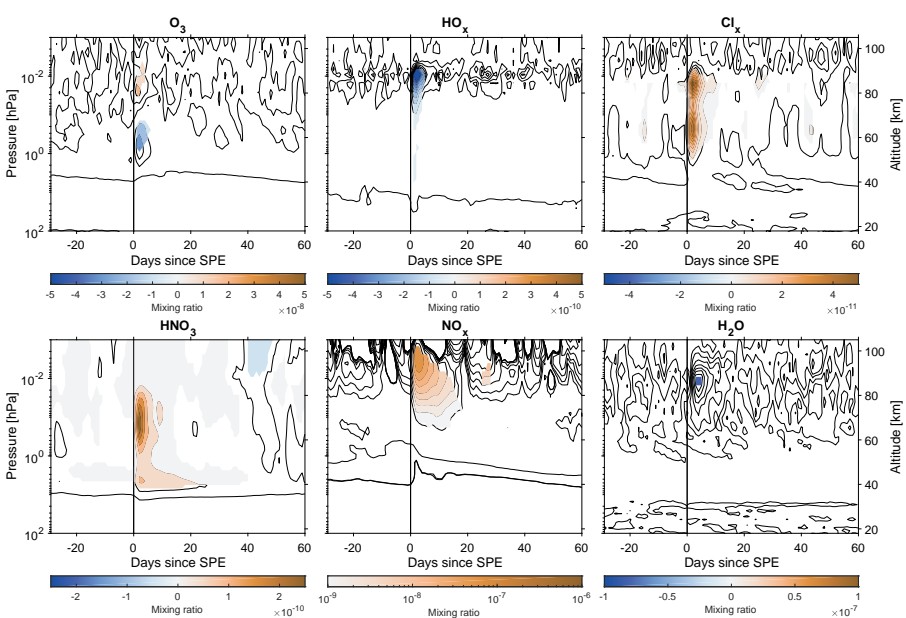

**Figure 7.** Area-weighted southern polar cap averages (latitude $> 60°$S) of composite means of $\widehat{WD}$ - $\widehat{REF}$ (see figure 5). X-axis shows the number of days before and after the event (day 0, solid black line) and y-axis pressure levels in model (hPa, left) and approximate altitude (km, right). Note the logarithmic color scale for $NO_x$.

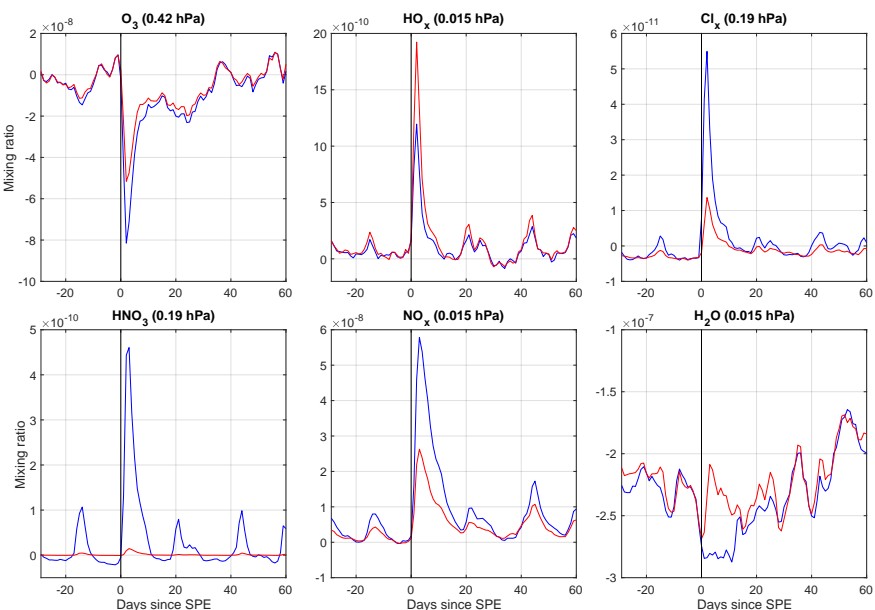

**Figure 8.** Composite mean $\widehat{WD}$ (blue) and $\widehat{REF}$ (red) for northern polar cap (area-weighted, latitude > 60°N) at selected pressure bands (three pressure levels, pressure of the center of the band shown in panel title). X-axis shows the number of days before and after the event (day 0, solid black line).

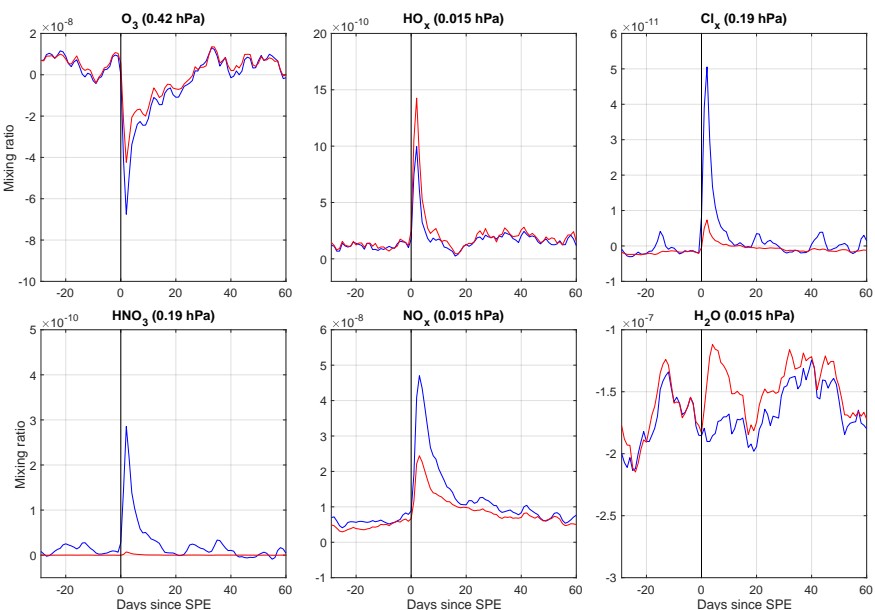

**Figure 9.** Composite mean $\widehat{WD}$ (blue) and $\widehat{REF}$ (red) for southern polar cap (area-weighted, latitude > 60°S) at selected pressure bands (three pressure levels, pressure of the center of the band shown in panel title). X-axis shows the number of days before and after the event (day 0, solid black line).



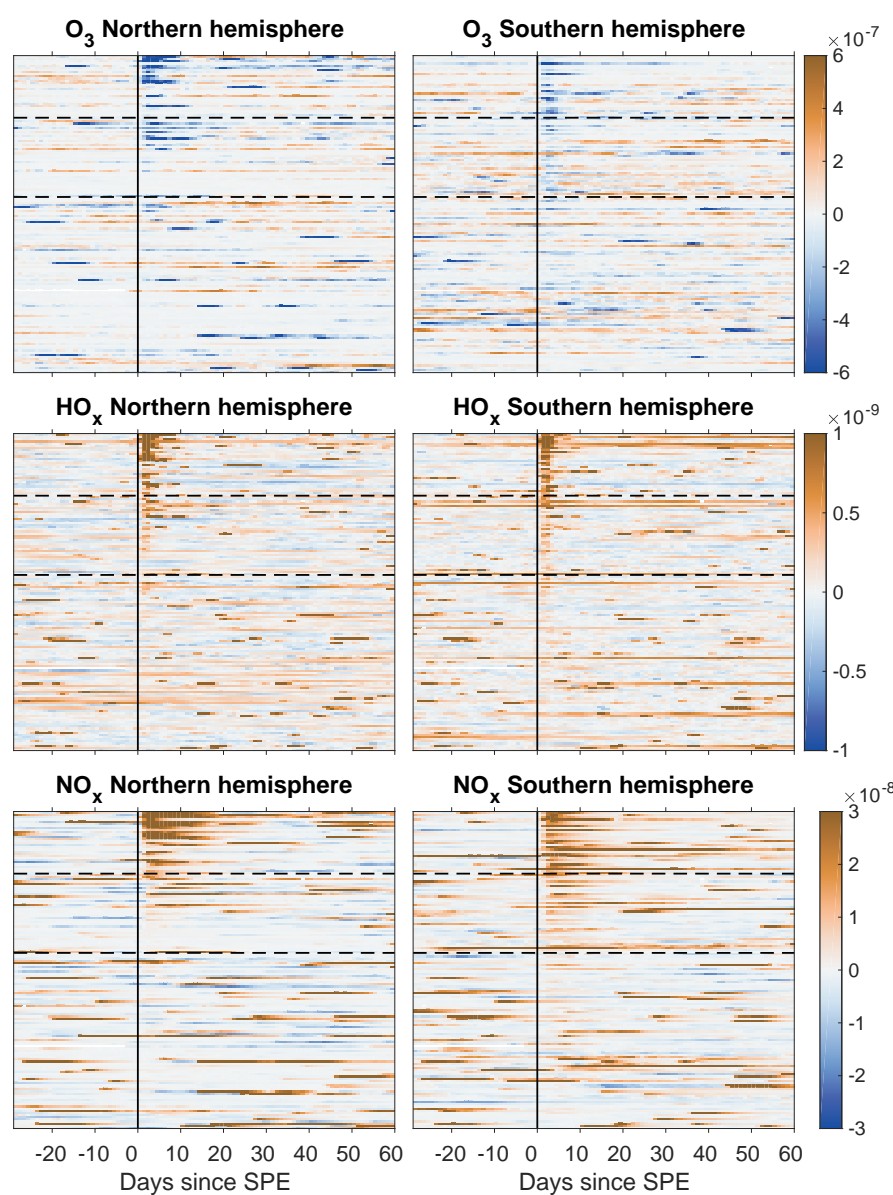

**Figure 10.** $\widehat{\text{WD}}$ of individual SPEs at 0.05 hPa pressure level, sorted by the increasing SPE magnitude for northern (left) and southern (right) polarcaps. Dashed lines represent the 100 and 1000 pfu limits.





**Table A1.** List of solar proton events used in analysis, with their start dates, dates of maximum intensity and the maximum proton fluxes (particle flux unit, $cm^{-2}s^{-1}sr^{-1}$).

| Start date | Maximum date | Proton Flux [pfu] | Start date | Maximum date | Proton Flux [pfu] |
|---|---|---|---|---|---|
| 08-Mar-1989 | 13-Mar-1989 | 3500 | 12-Sep-2000 | 13-Sep-2000 | 320 |
| 17-Mar-1989 | 18-Mar-1989 | 2000 | 08-Nov-2000 | 09-Nov-2000 | 14800 |
| 11-Apr-1989 | 12-Apr-1989 | 450 | 24-Nov-2000 | 26-Nov-2000 | 940 |
| 06-May-1989 | 06-May-1989 | 110 | 02-Apr-2001 | 03-Apr-2001 | 1110 |
| 12-Aug-1989 | 13-Aug-1989 | 9200 | 10-Apr-2001 | 11-Apr-2001 | 355 |
| 29-Sep-1989 | 30-Sep-1989 | 4500 | 15-Apr-2001 | 15-Apr-2001 | 951 |
| 19-Oct-1989 | 20-Oct-1989 | 40000 | 16-Aug-2001 | 16-Aug-2001 | 493 |
| 27-Nov-1989 | 28-Nov-1989 | 380 | 24-Sep-2001 | 25-Sep-2001 | 12900 |
| 30-Nov-1989 | 01-Dec-1989 | 7300 | 04-Nov-2001 | 06-Nov-2001 | 31700 |
| 19-Mar-1990 | 19-Mar-1990 | 950 | 22-Nov-2001 | 24-Nov-2001 | 18900 |
| 28-Apr-1990 | 28-Apr-1990 | 150 | 26-Dec-2001 | 26-Dec-2001 | 779 |
| 21-May-1990 | 22-May-1990 | 410 | 21-Apr-2002 | 21-Apr-2002 | 2520 |
| 01-Aug-1990 | 01-Aug-1990 | 230 | 22-May-2002 | 23-May-2002 | 820 |
| 31-Jan-1991 | 31-Jan-1991 | 240 | 16-Jul-2002 | 17-Jul-2002 | 234 |
| 23-Mar-1991 | 24-Mar-1991 | 43000 | 24-Aug-2002 | 24-Aug-2002 | 317 |
| 13-May-1991 | 13-May-1991 | 350 | 07-Sep-2002 | 07-Sep-2002 | 208 |
| 04-Jun-1991 | 11-Jun-1991 | 3000 | 09-Nov-2002 | 10-Nov-2002 | 404 |
| 14-Jun-1991 | 15-Jun-1991 | 1400 | 28-May-2003 | 29-May-2003 | 121 |
| 30-Jun-1991 | 02-Jul-1991 | 110 | 26-Oct-2003 | 26-Oct-2003 | 466 |
| 07-Jul-1991 | 08-Jul-1991 | 2300 | 28-Oct-2003 | 29-Oct-2003 | 29500 |
| 26-Aug-1991 | 27-Aug-1991 | 240 | 25-Jul-2004 | 26-Jul-2004 | 2086 |
| 09-May-1992 | 09-May-1992 | 4600 | 13-Sep-2004 | 14-Sep-2004 | 273 |
| 25-Jun-1992 | 26-Jun-1992 | 390 | 07-Nov-2004 | 08-Nov-2004 | 495 |
| 30-Oct-1992 | 31-Oct-1992 | 2700 | 16-Jan-2005 | 17-Jan-2005 | 5040 |
| 20-Feb-1994 | 21-Feb-1994 | 10000 | 14-May-2005 | 15-May-2005 | 3140 |
| 06-Nov-1997 | 07-Nov-1997 | 490 | 14-Jul-2005 | 15-Jul-2005 | 134 |
| 20-Apr-1998 | 21-Apr-1998 | 1700 | 22-Aug-2005 | 23-Aug-2005 | 330 |
| 02-May-1998 | 02-May-1998 | 150 | 08-Sep-2005 | 11-Sep-2005 | 1880 |
| 06-May-1998 | 06-May-1998 | 210 | 06-Dec-2006 | 07-Dec-2006 | 1980 |
| 24-Aug-1998 | 26-Aug-1998 | 670 | 23-Jan-2012 | 24-Jan-2012 | 6310 |
| 30-Sep-1998 | 01-Oct-1998 | 1200 | 07-Mar-2012 | 08-Mar-2012 | 6530 |
| 14-Nov-1998 | 14-Nov-1998 | 310 | 17-May-2012 | 17-May-2012 | 255 |
| 14-Jul-2000 | 15-Jul-2000 | 24000 | 17-Jul-2012 | 18-Jul-2012 | 136 |