# Peer review of "Statistical response of middle atmosphere composition to solar proton events in WACCM-D simulations: importance of lower ionospheric chemistry"

_Atmospheric Chemistry and Physics, 2019_

## Referee Comment (RC1) · Anonymous Referee #3 · 30 Jan 2020

General Comments: The authors use the WACCM-D model, a variant of the Whole Atmosphere Community Climate Model (WACCM), to study the statistical response of the atmosphere to the 66 largest solar proton events (SPEs) that occurred in years 1989-2012. WACCM-D, unlike the standard WACCM, includes a comprehensive ion chemistry set for the lower ionosphere with 307 reactions of 20 positive ions and 21 negative ions. Compared to the standard WACCM, WACCM-D produces a larger response in $O_3$ and $NO_x$, a weaker response in $HO_x$ and simulates changes in $HNO_3$ and $Cl_x$, which are in better agreement with observations. It is recommended that ion

chemistry reactions (similar to those in WACCM-D) be included in future models to study the impact of energetic particle precipitation (EPP) on the middle atmosphere. The article presents a good comparison of the WACCM-D versus standard WACCM results for the SPEs. The paper provides interesting results for the 66 largest SPEs in years 1989-2012. I do think that the paper should be published. The paper is generally well-written, but I have five specific comments and some suggested technical corrections/suggestions.

Specific Comments:

1) p. 5, line 23; p. 6, lines 6-10, mention of figures: Figure 3 is mentioned on p. 5 in line 23. The next four figures mentioned are Figures 6 and 7 in line 6 and Figures 8 and 9 in line 7. It is curious that Figures 4 and 5 are not mentioned until line 10 on p. 6 of the text. I assume that the two figures (4 & 5) will be positioned in the manuscript right after Figure 3. It is certainly reasonable that Figures 4 & 5 be positioned before Figures 6, 7, 8, and 9. Therefore, it is suggested that Figures 4 & 5 be mentioned in the text between line 23 on p. 5 and line 6 on p. 6.

2) p. 16, Figure 3: Unless the two plots are repositioned, currently "(top)" should be changed to "(left)" and "(bottom)" should be changed to "(right)" in the figure caption.

3) pp. 17-20, Figures 4-7: It is unclear what intervals and mixing ratio values the contour lines illustrate. There are no numbers associated with the contour levels. Possibly remove the contour lines as their significance is vague. The colors in the figures are fairly clear.

4) p. 23, Figure 10: It might be helpful to label the 100 and 1000 pfu dashed lines on the three far left y-axes as "1000 pfu" and "100 pfu."

5) p. 24, Table A1: The date with the largest Proton Flux (pfu) of 43000 has a Start date "23-Mar-1991" and a Maximum date "24-Mar-1991." I was surprised to see that this solar proton event (SPE) had the largest Proton Flux, as I have not read or heard

much about this particular SPE. Perhaps some research of the measured atmospheric impact of this very large SPE is needed in a future study.

Technical Corrections/Suggestions:

1) p. 1, line 13: Change "weaker response" to "a weaker response"

2) p. 2, line 17: Change "chemistry which connects SPE ionization to changes in neutral species" to "chemistry, which connects SPE ionization to changes in neutral species,"

3) p. 2, line 28: Change "Difference" to "The difference"

4) p. 3, line 11: Change "of number" to "of a number"

5). 3, line 21: Change "in long-term" to "in a long-term"

6) p. 4, line 30: Change "underpresented" to "underrepresented"

7) p. 5, line 6: Change "been seen" to "be seen"

8) p. 5, line 21: Change "have also" to "also have"

9) p. 7, line 4: Change "is a reduced" to "is reduced"

10) p. 7, line 7: Change "with short-lived" to "with a short-lived"

11) p. 7, line 11: Change "in mesosphere" to "in the mesosphere"

12) p. 7, line 13: Change "Large increase" to "A large increase"

13) p. 7, line 15: Change "Longer-term" to "A longer-term"

14) p. 7, line 20: Change "Response" to "The response"

15) p. 8, line 8: Change "clear no connection" to "no clear connection"

16) p. 8, line 13: Change "reason" to "reasons"

17) p. 9, line 9: Change "astronger" to "a stronger"

18) p. 9, line 12: Change "for weakest" to "for the weakest"

19) p. 9, line 19: Change "from detailed" to "from a detailed"

20) p. 10, line 1: Change "due the less" to "due to less"

---

## Referee Comment (RC2) · Niilo Kalakoski et al. · 14 Feb 2020

This paper investigates the impact of explicitly including D-region ion chemistry instead of simple parameterizations in a global model, on the chemical composition of the middle atmosphere during and after large solar proton events (SPEs). This is investigated by comparing results from a model run over the period 1989-2012 using full D-region ion chemistry with a model using the standard parameterizations producing NOx and HOx as a function of the ion pair production rate. A clear impact is shown on the amount of NOx and HNO3 produced during the event, as well as on active chlorine Clx. Ozone is affected mainly around the stratopause, presumably due to the additional Clx avail-

able. As energetic particle precipitation from SPEs and the aurora are considered part of the solar forcing of the climate system, this is an interesting and important result in terms of understanding the response of the chemical composition to atmospheric ionization. The paper is well written and to the point, and I recommend publication after addressing a few mostly technical points listed below. One point I want to emphasize here which should be added to the discussion of the results: As the impact of using the full D-region ion chemistry instead of simple parameterizations on ozone seems to be small and restricted mainly to the stratopause, the simple parameterizations are therefore likely sufficient for long climate projections including the particle precipitation contribution to top-down solar forcing.

Page 1, line 3: "SPEs cause production of odd hydrogen and odd nitrogen" better maybe "odd hydrogen and odd nitrogen are produced during SPEs"

Page 1, line 4: "the largest events" –> "the strongest events"

Page 1, line 9: ... to the 66 "strongest" SPEs "which" occurred in "the" years ...

Page 1, Introduction, first sentence: I found this explanation of the nature of SPEs too vague, particularly considering the source of the high-energy protons. Maybe better: "Solar proton events are observed on Earth when high-energy protons accelerated in the sun's magnetic field during a solar coronal mass ejection strike Earth."

Page 1, line 18: ... at "magnetic" (or geomagnetic?) latitudes "polewards of" $\sim 60°$.

Page 1, line 18: This causes "excitation", ionization and dissociation ... the excitation is often forgotten in this context, but these are of course what forms the visible aurora or polar cap absorption related to geomagnetic activity and even SPEs, and e.g., N(2D) and O(1D) are actually very important for the response of the chemical composition.

Page 2, line 6: there are much more papers investigating HOx and NOx production and ozone loss during and after SPEs (including a fairly long list by Charlie Jackman starting 1980). I appreciate you don't need to list them all, but maybe add "e.g., " before

the references to emphasize that this is just a selection?

Page 2, line 8: Rozanov et al 2005 did not include SPEs, they only considered an upper boundary NOx source. You could cite Rozanov et al, Surv. Geophys., 2012 - they did include SPEs, upper boundary, and GCRs. Likewise Baumgaertner et al 2011 only included an upper boundary NOx source at the top of his model (0.01 hPa, about 80 km), so definitely did not consider SPEs. Seppaelae et al did not exclude SPEs, so her "high geomagnetic activity" probably was biased to SPE years. Even if your statement – SPEs as part of EEP can modulate winter dynamics – is very general, I think you should reference only studies that actually included SPEs here.

Page 2, line 10: the impact of energetic particle precipitation, and in particular SPEs and geomagnetic forcing, on the variability of stratospheric ozone has been discussed in the recent WMO assessment: WMO (World Meteorological Organization), Scientific Assessment of Ozone Depletion: 2018, Global Ozone Research and Monitoring Project – Report No. 58, 588 pp., Geneva, Switzerland, 2018.available online at www.esrl.noaa.gov/csd/assessments/ozone/2018/. This summarizes the state of the art, and should be references here as well.

Page 2, line 17: erase the i.e.

Page 2, line 32: the correct spelling is "von Savigny", no capital on the von.

Page 3, lines 8-9, O3, HOx, NOx "and HNO3"

Page 3, line 23: does the chemistry code include excited species like N(2D), O(1D), O(1S), O2(1Delta), O2(1Sigma) ...? Please add.

Page 4, line 30: underpresented –> did you mean "underrepresented"?

Page 6, line 17: These are most notable in "NOx and HNO3" in NH around ...

Page 6, line 27: ... highest energy protons (E>300MeV) "which" can ...

Page 6, line 31: ... with strongest "and most significant" response ...

Page 7, line 4: This decrease is roughly consistent ... and/or continuous?

Page 8, line 6: 0.5-1 hPa, I would say

Page 8, line 16: Secondary enhancements around day 40 are clearly visible in Clx.

Page 8, line 21: Same for Clx, see comment above.

Page 9, line 9: blank missing in astronger

Page 10, end of conclusion: However, as O3 loss in the stratosphere below 1hPa is not affected significantly, this will likely not have an impact on stratospheric dynamics and possible downward coupling to tropospheric weather systems. This means that in climate projections considering particle impacts as part of the solar forcing, ion chemistry probably does not need to be included.
* * *

---

## Referee Comment (RC3) · Anonymous Referee #1 · 14 Feb 2020

This paper analyses the impact of D-region ion chemistry on the middle atmospheric composition responses to solar proton events by means of superposed epoch analysis of standard WACCM and WACCM-D simulations covering 1989-2012. The authors identify important differences of simulated responses for $NO_x$, $O_3$, $HO_x$, and $Cl_x$, highlighting the importance of including ion chemistry reactions in models used to study EPP. This is a relevant result, particularly when considering that EPP is increasingly considered in climate models as a part of the solar forcing. The paper is written in a clear and concise manner. Overall, I recommend publication after addressing my

comments below, most of them being minor.

General comments:

1) I would have liked to see an analysis separating for seasons instead of (or in addition to) the analysis for SH and NH. This is motivated by the strong dependence of the SPE responses on the prevailing illumination and dynamical conditions (i.e., photochemistry, polar winter transport etc.), being particularly relevant for chlorine responses. I understand that the main purpose of the paper is to identify the impact of explicit D-region ion chemistry - being probably less affected by seasonal impacts (though much of the discussion is dedicated to gas phase chemistry impacts) . I'm further aware that such analysis implies additional problems (e.g., different ionisation levels during different seasons due to the uneven distributions of SPEs). In this sense, I'm not insisting in such additional analysis. However, if not included, the authors should at least be more quantitative about the prevailing seasonal conditions in the NH/SH. It is not sufficient to only mention that "the strongest SPEs occurred during NH winter".

2) It is very difficult to compare the presented results quantitatively with other studies dealing with individual events. This could easily remedied by providing a number that expresses the epoch ionisation level as fraction of that of a well-studied event such as the Halloween SPE.

Specific comments:

p1 l18: geomagnetic latitudes above 60 deg

p2 l8-9: none of the cited studies deals with SPE impacts.

p2 l11-16: This paragraph is confusing as it mixes up the different aspects "direct vs indirect ozone impacts" and "depletion by HOx/NOx chemistry vs increases due to chlorine buffering". I recommend to reorder this paragraph in the following manner: 1) simulated TOC decreases (Jackman et al. 2014) and reported local depletions in the lower stratosphere (Denton et al. 2018) 2) Lower stratospheric decreases are indirect

effects (Jackman et al, 2011) 3) On the other hand, local chemical impacts (chlorine buffering) may compensate indirect effects in the lower stratosphere (Jackman et. al., 2008).

p2 l28: Differences

p3 l2: . . .led to AN improved. . .

p3 l12: . . .to A number of. . .

p3 l12-13: The responses are not studied here in dependence of background atmosphere or illumination. In this sense, the statistical analysis performed here allows only for an evaluation of the CLIMATOLOGICAL response. Also, the "timing" (l14) is not studied explicitly in this paper.

p6 l24: Interestingly, there are apparently SPE-related short-term increases in the NH (not visible in the SH) which, however, occur slightly BEFORE the SPE onset. Any explanation?

p6 l25: Since 5hPa ozone responses are seen throughout the epoch period they are likely not caused by SPE. Instead they might be related to UV-induced solar cycle effects.

p6 l26: It might be possible that TOC decreases as reported by Denton et al. would be visible in an analysis restricted to polar winter (as done in the cited study).

p7 l2: Isn't the lowering of the peak altitude related to the increasingly (with altitude) smaller availability of water vapour during solar maximum conditions? How can HOx then increase at below 0.001 hPa? Please explain!

p7 l7-8: Short-term decreases (during nighttime conditions) as observed in 2005 in the NH are likely caused by conversion of ClO to HOCl in the presence of enhanced HO2 (see e.g. Funke et al., 2011). An increase after 30 days, as seen here, is not a short-term increase! A more plausible explanation - at least for the SH negative anomaly

[Figure]

- appears to be the descent of NOx and subsequent formation of ClONO2 (note that the NOx contours at 5 hPa seem to decrease with time more pronounced in the SH compared to the NH).

p7 l12: Isn't it more relevant in this context that HOCl is converted to ClO during DAY-TIME?

p7 l16: While the few strongest SPEs occurred in NH (early winter), it is not easy to infer if this is also true for the epoch average.

p7 l20: THE response

p7 l22: This is not easy to infer from Fig. 4. Do the contours within the white areas (with low significance) indicate decreases or increases? If they indicate decreases, then it would look more like the NH events being more short-lived compared to the SH events.

p7 l24: If winter conditions - allowing for descent of thermospheric NOx - were prevailing in the NH, shouldn't the anomaly be more pronounced there compared to the SH?

p8 l8-9: Why only in the SH? reduced HOx responses in WACCM D occur in both hemispheres. Could it be that, in the NH, Clx increases outweigh HOx increases in contrast to the SH?

p9 l9: a stronger

---

## Author Comment (AC1) · 8 May 2020

**Answers to comments by anonymous referees to manuscript acp-2019-1133: Kalakoski et al., Statistical response of middle atmosphere composition to solar proton events in WACCM-D simulations: importance of lower ionospheric chemistry**

Authors would like to thank the referees for their time and comments. Please find below our answers (in blue) to the comments (in black).

[Figure]

Please also note one additional issue requiring correction in the original manuscript. Meteorological forcing fields in the model come from Modern-Era Retrospective Analysis for Research and Applications (MERRA), not from the GEOS 5.1 as it was written in the original text (p. 4 Lines 7-8). In the revised manuscript, the words "from NASA GMAO GEOS5.1 (Reinecker et al., 2008)" have been substituted by "from NASA's Modern-Era Retrospective Analysis for Research and Applications (MERRA) (Lamarque et al., 2012, Rienecker et al., 2011)", with appropriate references. Note that this mistake does not affect our results or conclusions.

**Anonymous Referee 1**

This paper analyses the impact of D-region ion chemistry on the middle atmospheric composition responses to solar proton events by means of superposed epoch analysis of standard WACCM and WACCM-D simulations covering 1989-2012. The authors identify important differences of simulated responses for NOx, O3, HOx, and Clx, highlighting the importance of including ion chemistry reactions in models used to study EPP. This is a relevant result, particularly when considering that EPP is increasingly considered in climate models as a part of the solar forcing. The paper is written in a clear and concise manner. Overall, I recommend publication after addressing my comments below, most of them being minor.

General comments: 1) I would have liked to see an analysis separating for seasons instead of (or in addition to) the analysis for SH and NH. This is motivated by the strong dependence of the SPE responses on the prevailing illumination and dynamical conditions (i.e., photochemistry, polar winter transport etc.), being particularly relevant for chlorine responses. I understand that the main purpose of the

paper is to identify the impact of explicit D- region ion chemistry - being probably less affected by seasonal impacts (though much of the discussion is dedicated to gas phase chemistry impacts) . I'm further aware that such analysis implies additional problems (e.g., different ionisation levels during different seasons due to the uneven distributions of SPEs). In this sense, I'm not insisting in such additional analysis. However, if not included, the authors should at least be more quantitative about the prevailing seasonal conditions in the NH/SH. It is not sufficient to only mention that "the strongest SPEs occurred during NH winter".

As the referee pointed out, the different ionization levels and dynamical conditions in different seasons made the analysis more ambiguous when the events were divided by seasons. Also, the better statistics obtained by analysing the full year as a single dataset was deemed more advantageous for the analysis as presented.

However, we agree that additional clarification should be added to text describing the distribution of the SPEs in the simulation.

New text is added to the section 2.2, replacing the last sentence in paragraph on p.4, lines 30-31:

Of the 8 largest SPEs (above 10000 pfu) in the analysis, five occurred October or November, compared to only each in March, July and September.

2) It is very difficult to compare the presented results quantitatively with other studies dealing with individual events. This could easily remedied by providing a number that expresses the epoch ionisation level as fraction of that of a well-studied event such as the Halloween SPE.

We agree, and have now added such a comparison to the text.

Daily composite mean peak ionisation is 891 ion pairs/cm3/s, which is similar in magnitude to peak ionisation associated with January 2005 SPE. Largest events in the time series feature peak ionisations about ten times higher, for example peak ionisation of 8534 ion pairs/cm3/s for the Halloween SPE.

Specific comments:

p2 l8-9: none of the cited studies deals with SPE impacts.

References are modified accordingly. References not dealing with SPEs were removed. As referee 2 pointed out, Seppälä et al. is likely biased to SPE years so we choose to retain that reference. Additional references (Semeniuk et al., 2011, Rozanov et al., 2012, Calisto et al., 2013), including SPEs, were added.

Semeniuk, K., Fomichev, V. I., McConnell, J. C., Fu, C., Melo, S. M. L., and Usoskin, I. G. (2011). Middle atmosphere response to the solar cycle in irradiance and ionizing particle precipitation. Atmospheric Chemistry and Physics, 11(10), 5045.

Rozanov, E., Calisto, M., Egorova, T., Peter, T., and Schmutz, W. (2012). Influence of the precipitating energetic particles on atmospheric chemistry and climate. Surveys in geophysics, 33(3-4), 483-501.

Calisto, M., Usoskin, I., and Rozanov, E. (2013). Influence of a Carrington-like event

on the atmospheric chemistry, temperature and dynamics: revised. Environmental Research Letters, 8(4), 045010.

p2 l11-16: This paragraph is confusing as it mixes up the different aspects "direct vs indirect ozone impacts" and "depletion by HOx/NOx chemistry vs increases due to chlorine buffering". I recommend to reorder this paragraph in the following manner: 1) simulated TOC decreases (Jackman et al. 2014) and reported local depletions in the lower stratosphere (Denton et al. 2018) 2) Lower stratospheric decreases are indirect effects (Jackman et al, 2011) 3) On the other hand, local chemical impacts (chlorine buffering) may compensate indirect effects in the lower stratosphere (Jackman et. al., 2008).

We have revised the text as suggested. The new text is:

Simulation results have suggested that the decrease of polar total ozone column would be of the order of 1–2% a few months after large SPEs (Jackman et al., 2014). However, local depletion has been reported to reach ≈10% below the ozone layer peak at 50–100 hPa, based on a statistical analysis of almost 200 SPEs using ozone soundings (Denton et al., 2018). As the contribution of >300 MeV protons to direct ozone loss in the lower stratosphere would likely be negligible due to the relatively small fluxes at such high energies (Jackman et al., 2011), any observed ozone loss would likely be the result of indirect effects. Ozone may also increase in the lower stratosphere due to the enhanced NOx interfering with chlorine-driven catalytic ozone loss (Jackman et al., 2008).

p3 l12-13: The responses are not studied here in dependence of background atmosphere or illumination. In this sense, the statistical analysis performed here
allows only for an evaluation of the CLIMATOLOGICAL response. Also, the "timing" (l14) is not studied explicitly in this paper.

We have revised the text for clarity, taking into account the comments. The new text is: This approach also allows for identification of climatological effects above natural variability. As the analysis includes SPEs of different sizes occuring during different seasons, a statistical approach is most useful for study of temporal and spatial extent, rather than magnitude of the response.

p6 l24: Interestingly, there are apparently SPE-related short-term increases in the NH (not visible in the SH) which, however, occur slightly BEFORE the SPE onset. Any explanation?

It is likely that O3 increase above 10-2 hPa in NH is similar to the solar cycle signal that is observed in SH, just less robust statistically and thus more patched in the figure.

p6 l25: Since 5hPa ozone responses are seen throughout the epoch period they are likely not caused by SPE. Instead they might be related to UV-induced solar cycle effects.

Yes, that was our intention in this sentence. We have revised the text for clarity. New text is: In the stratosphere below 5 hPa, the ozone response is not consistently robust. However, an intermittent increase, likely caused by solar cycle effects, is seen throughout the epoch period in this region.

p6 l26: It might be possible that TOC decreases as reported by Denton et al. would be visible in an analysis restricted to polar winter (as done in the cited study).

Our preliminary study of the seasonally separated events did not reveal such an effect in winter. However, follow-up concentrating on winter events is certainly needed. We have looked at the seasonal effects more closely in a separate paper (Jia et al., 2020, submitted to ACP). However, no TOC decreases were found in the follow-up study using WACCM-D simulations or satellite observations.

p7 l2: Isn't the lowering of the peak altitude related to the increasingly (with altitude) smaller availability of water vapour during solar maximum conditions? How can HOx then increase at below 0.001 hPa? Please explain!

This is likely due to increased Lyman-alpha penetrating deeper in the atmosphere during solar maxima and producing more OH below the usual peak altitude, thus counteracting the decrease of H2O.

New text, added after p7, l2: Factors affecting HOx peak a(ltitude during solar maxima are the decrease of H2O and the increase of Lyman-Alpha photolysis, balancing at the level of mesopause.

p7 l7-8: Short-term decreases (during nighttime conditions) as observed in 2005 in the NH are likely caused by conversion of ClO to HOCl in the presence of enhanced HO2 (see e.g. Funke et al., 2011). An increase after 30 days, as seen here, is not a short- term increase! A more plausible explanation - at least for the SH negative anomaly - appears to be the descent of NOx and subsequent formation of ClONO2 (note that the NOx contours at 5 hPa seem to decrease with time more pronounced in

the SH compared to the NH).

Agreed, manuscript was revised accordingly. New text: Short-lived decrease of the NH polar upper stratosphere ClO during the January 2005 SPE has previously been reported both in satellite observations and models (Damiani et al., 2012; Andersson et al., 2016). In short term decrease is likely caused by conversion of ClO to HOCl in the presence of enhanced HO2, while in the longer term, the decrease is probably connected to descent of NOx and subsequent formation of ClONO2.

p7 l12: Isn't it more relevant in this context that HOCl is converted to ClO during DAYTIME?

Manuscript was revised accordingly. New text: ".., while in daytime HOCl is converted to ClO by OH."

p7 l16: While the few strongest SPEs occurred in NH (early winter), it is not easy to infer if this is also true for the epoch average.

More quantitative analysis of SPE numbers was added to section 2.2 (see answer to general comment 1).

The importance of the amount of available sunlight for HNO3 is seen in figure in supplement (analogous to figure 10 in the manuscript). Even for the largest events (top lines in each panel), large increase is only observed where the event occurs during the polar night for that hemisphere.

p7 l22: This is not easy to infer from Fig. 4. Do the contours within the white areas

(with low significance) indicate decreases or increases? If they indicate decreases, then it would look more like the NH events being more short-lived compared to the SH events.

NOx panels of the figures 4 and 5 are in logarithmic scale, so all contours indicate an increase. In NH, this increase is however markedly lower between events, which we interpreted here as a more consistent response in SH due to less variability in wintertime dynamics. We agree that additional explanation to NOx panels should be included.

New text, replacing the last sentence of captions of figures 4-7: Note that the color scale for NOx panel is logarithmic, and all contours shown in that panel indicate positive difference.

p7 l24: If winter conditions - allowing for descent of thermospheric NOx - were prevailing in the NH, shouldn't the anomaly be more pronounced there compared to the SH?

We speculate this is due to more stable dynamics in SH which allow the anomalies to last longer. Although the mean effect actually is stronger in NH (figures 7 and 8), SH response is less transient.

p8 l8-9: Why only in the SH? reduced HOx responses in WACCM D occur in both hemispheres. Could it be that, in the NH, Clx increases outweigh HOx increases in contrast to the SH?

The difference in plots in figures is due to the difference in natural variability between

hemispheres. Positive anomaly is of similar magnitude in NH, it is just statistically less robust than in SH.

New text: Less ozone depletion takes place in WACCM-D around 0.01 hPa level, connected to the clearly lower HOx enhancement in WACCM-D. Anomaly of similar magnitude is observed in both hemispheres, although it is only robust in SH.

p1 l18: geomagnetic latitudes above 60 deg
p2 l28: Differences
p3 l2: ...led to AN improved... p3 l12: ...to A number of...
p7 l20: THE response
p9 l9: a stronger

Manuscript is updated accordingly.

**Anonymous Referee 2**

This paper investigates the impact of explicitly including D-region ion chemistry instead of simple parameterizations in a global model, on the chemical composition of the middle atmosphere during and after large solar proton events (SPEs). This is investigated by comparing results from a model run over the period 1989-2012 using full D-region ion chemistry with a model using the standard parameterizations producing NOx and HOx as a function of the ion pair production rate. A clear impact is shown on the amount of NOx and HNO3 produced during the event, as well as on active chlorine Clx. Ozone is affected mainly around the stratopause, presumably due to the additional Clx available. As energetic particle precipitation from SPEs and the aurora are considered part of the solar forcing of the climate system, this is an

interesting and important result in terms of understanding the response of the chemical composition to atmospheric ionization. The paper is well written and to the point, and I recommend publication after addressing a few mostly technical points listed below. One point I want to emphasize here which should be added to the discussion of the results: As the impact of using the full D-region ion chemistry instead of simple parameterizations on ozone seems to be small and restricted mainly to the stratopause, the simple parameterizations are therefore likely sufficient for long climate projections including the particle precipitation contribution to top-down solar forcing.

Thank you for your positive comments. Your point, re. interpretation of the importance of D-region chemistry is discussed below.

Page 1, Introduction, first sentence: I found this explanation of the nature of SPEs too vague, particularly considering the source of the high-energy protons. Maybe better: "Solar proton events are observed on Earth when high-energy protons accelerated in the sun's magnetic field during a solar coronal mass ejection strike Earth."

We agree, the text has been revised as suggested.

Page 1, line 18: ... at "magnetic" (or geomagnetic?) latitudes "polewards of" 60 .

Manuscript has been updated accordingly, see also comment form Referee 1.

Page 1, line 18: This causes "excitation", ionization and dissociation ... the excitation is often forgotten in this context, but these are of course what forms the visible aurora

or polar cap absorption related to geomagnetic activity and even SPEs, and e.g., N(2D) and O(1D) are actually very important for the response of the chemical composition.

Excitation was added to the sentence.

Page 2, line 6: there are much more papers investigating HOx and NOx production and ozone loss during and after SPEs (including a fairly long list by Charlie Jackman starting 1980). I appreciate you don't need to list them all, but maybe add "e.g., " before the references to emphasize that this is just a selection?

Agreed, modified accordingly.

Page 2, line 8: Rozanov et al 2005 did not include SPEs, they only considered an upper boundary NOx source. You could cite Rozanov et al, Surv. Geophys., 2012 - they did include SPEs, upper boundary, and GCRs. Likewise Baumgaertner et al 2011 only included an upper boundary NOx source at the top of his model (0.01 hPa, about 80 km), so definitely did not consider SPEs. Seppaelae et al did not exclude SPEs, so her "high geomagnetic activity" probably was biased to SPE years. Even if your statement – SPEs as part of EEP can modulate winter dynamics – is very general, I think you should reference only studies that actually included SPEs here.

Agreed, references are modified accordingly. Rozanov et al 2005 and Baumgartner 2011 were removed. Seppälä et al. is retained and additional references (Semeniuk et al., 2011, Rozanov et al., 2012, Calisto et al., 2013), including SPEs, added.

Semeniuk, K., Fomichev, V. I., McConnell, J. C., Fu, C., Melo, S. M. L., and Usoskin, I. G. (2011). Middle atmosphere response to the solar cycle in irradiance and ionizing particle precipitation. Atmospheric Chemistry and Physics, 11(10), 5045.

Rozanov, E., Calisto, M., Egorova, T., Peter, T., and Schmutz, W. (2012). Influence of the precipitating energetic particles on atmospheric chemistry and climate. Surveys in geophysics, 33(3-4), 483-501.

Calisto, M., Usoskin, I., and Rozanov, E. (2013). Influence of a Carrington-like event on the atmospheric chemistry, temperature and dynamics: revised. Environmental Research Letters, 8(4), 045010.

Page 2, line 10: the impact of energetic particle precipitation, and in particular SPEs and geomagnetic forcing, on the variability of stratospheric ozone has been discussed in the recent WMO assessment: WMO (World Meteorological Organization), Scientific Assessment of Ozone Depletion: 2018, Global Ozone Research and Monitoring Project – Report No. 58, 588 pp., Geneva, Switzerland, 2018. Available online at www.esrl.noaa.gov/csd/assessments/ozone/2018/. This summarizes the state of the art, and should be references here as well.

Agreed, we added the reference and following new text (after p.2 l.10): State of art on the impact of energetic particle precipitation, in particular SPE and geomagnetic forcing, has also been summarized in the recent WMO assessment (WMO, 2018, chapter 4.3.5.1, page 4-25, and references therein).

Page 3, line 23: does the chemistry code include excited species like N(2D), O(1D), O(1S), O2(1Delta), O2(1Sigma) ...? Please add.

Excited species N(2D), O(1D), O2(1Delta) and O2(1Sigma) are included in the standard WACCM, and by extension in WACCM-D. Added to the description.

Page 7, line 4: This decrease is roughly consistent ... and/or continuous?

Yes, that might be clearer. Will update accordingly.

Page 8, line 16: Secondary enhancements around day 40 are clearly visible in Clx.
Page 8, line 21: Same for Clx, see comment above.

Agree, line to that effect was added.

New text, p.8, l.15: Enhancements relating to secondary ionization peaks can also be seen, most clearly around day 40.

Page 10, end of conclusion: However, as O3 loss in the stratosphere below 1hPa is not affected significantly, this will likely not have an impact on stratospheric dynamics and possible downward coupling to tropospheric weather systems. This means that in climate projections considering particle impacts as part of the solar forcing, ion chemistry probably does not need to be included.

We agree that based on this study, the addition of D-region chemistry does not lead to statistically robust, additional ozone response in the stratosphere. However, here we look only at the 60-day effect after sporadic SPEs, while in the case of longer-term forcing, e.g. from MEE, the situation might be different. Note also that our simulations

have increased mesospheric NOx when ion chemistry is included, which in longer time scales would descend to the upper stratosphere and affect ozone.

Page 1, line 3: "SPEs cause production of odd hydrogen and odd nitrogen" better maybe "odd hydrogen and odd nitrogen are produced during SPEs"
Page 1, line 4: "the largest events" –> "the strongest events"
Page 1, line 9: ... to the 66 "strongest" SPEs "which" occurred in "the" years . . .
Page 2, line 17: erase the i.e.
Page 2, line 32: the correct spelling is "von Savigny", no capital on the von.
Page 3, lines 8-9, O3, HOx, NOx "and HNO3"
Page 4, line 30: underpresented –> did you mean "underrepresented"?
Page 6, line 17: These are most notable in "NOx and HNO3" in NH around ... Page 6, line 27: ... highest energy protons (E>300MeV) "which" can ...
Page 6, line 31: ... with strongest "and most significant" response ...
Page 8, line 6: 0.5-1 hPa, I would say
Page 9, line 9: blank missing in astronger

Manuscript was updated accordingly.

**Anonymous Referee 3**

General Comments: The authors use the WACCM-D model, a variant of the Whole Atmosphere Community Climate Model (WACCM), to study the statistical response of the atmosphere to the 66 largest solar proton events (SPEs) that occurred in years 1989-2012. WACCM-D, unlike the standard WACCM, includes a comprehensive ion chemistry set for the lower ionosphere with 307 reactions of 20 positive ions and 21 negative ions. Compared to the standard WACCM, WACCM-D produces a larger

response in O3 and NOx, a weaker response in HOx and simulates changes in HNO3 and Clx, which are in better agreement with observations. It is recommended that ion chemistry reactions (similar to those in WACCM-D) be included in future models to study the impact of energetic particle precipitation (EPP) on the middle atmosphere. The article presents a good comparison of the WACCM-D versus standard WACCM results for the SPEs. The paper provides interesting results for the 66 largest SPEs in years 1989-2012. I do think that the paper should be published. The paper is generally well-written, but I have five specific comments and some suggested technical corrections/suggestions.

Specific Comments:

1) p. 5, line 23; p. 6, lines 6-10, mention of figures: Figure 3 is mentioned on p. 5 in line 23. The next four figures mentioned are Figures 6 and 7 in line 6 and Figures 8 and 9 in line 7. It is curious that Figures 4 and 5 are not mentioned until line 10 on p. 6 of the text. I assume that the two figures (4 and 5) will be positioned in the manuscript right after Figure 3. It is certainly reasonable that Figures 4 and 5 be positioned before Figures 6, 7, 8, and 9. Therefore, it is suggested that Figures 4 and 5 be mentioned in the text between line 23 on p. 5 and line 6 on p. 6.

Agreed, figures 4 and 5 are now introduced in p. 6 line 5.

2) p. 16, Figure 3: Unless the two plots are repositioned, currently "(top)" should be changed to "(left)" and "(bottom)" should be changed to "(right)" in the figure caption.

Figure was repositioned, but the caption was not. Manuscript was corrected accordingly.

3) pp. 17-20, Figures 4-7: It is unclear what intervals and mixing ratio values the contour lines illustrate. There are no numbers associated with the contour levels. Possibly remove the contour lines as their significance is vague. The colors in the figures are fairly clear.

We agree that the significance of the contour lines might not be clear in every case. However, in some cases the non-robust signals are discussed in the text, and we would prefer to keep the contour lines in all figures for consistency.

4) p. 23, Figure 10: It might be helpful to label the 100 and 1000 pfu dashed lines on the three far left y-axes as "1000 pfu" and "100 pfu."

Agreed, meaning of the dashed lines should be clarified. "(top)" and "(bottom)" qualifiers were added to the caption.

5) p. 24, Table A1: The date with the largest Proton Flux (pfu) of 43000 has a Start date "23-Mar-1991" and a Maximum date "24-Mar-1991." I was surprised to see that this solar proton event (SPE) had the largest Proton Flux, as I have not read or heard much about this particular SPE. Perhaps some research of the measured atmospheric impact of this very large SPE is needed in a future study.

It is true that the impact of the March 1991 Solar storm on the atmosphere seems to be less studied than later large events. Partially this is due to the wealth of satellite observations from instruments like GOMOS, MIPAS and SCIAMACHY available for the later events. Energy spectrum of the March 1991 event was apparently fairly soft,

with no detectable ground level event.

Technical Corrections/Suggestions:
1) p. 1, line 13: Change "weaker response" to "a weaker response"
2) p. 2, line 17: Change "chemistry which connects SPE ionization to changes in neutral species" to "chemistry, which connects SPE ionization to changes in neutral species,"
3) p. 2, line 28: Change "Difference" to "The difference"
4) p. 3, line 11: Change "of number" to "of a number"
5). 3, line 21: Change "in long-term" to "in a long-term"
6) p. 4, line 30: Change "underpresented" to "underrepresented"
7) p. 5, line 6: Change "been seen" to "be seen"
8) p. 5, line 21: Change "have also" to "also have"
9) p. 7, line 4: Change "is a reduced" to "is reduced"
10) p. 7, line 7: Change "with short-lived" to "with a short-lived"
11) p. 7, line 11: Change "in mesosphere" to "in the mesosphere"
12) p. 7, line 13: Change "Large increase" to "A large increase"
13) p. 7, line 15: Change "Longer-term" to "A longer-term"
14) p. 7, line 20: Change "Response" to "The response"
15) p. 8, line 8: Change "clear no connection" to "no clear connection"
16) p. 8, line 13: Change "reason" to "reasons"
17) p. 9, line 9: Change "astronger" to "a stronger"
18) p. 9, line 12: Change "for weakest" to "for the weakest"
19) p. 9, line 19: Change "from detailed" to "from a detailed"
20) p. 10, line 1: Change "due the less" to "due to less"

Manuscript was modified accordingly.

**Supplement:**

**HNO₃ Northern hemisphere**

**HNO₃ Southern hemisphere**